# Antioxidant Scavenging of the Superoxide Radical by Yerba Mate (*Ilex paraguariensis)* and Black Tea *(Camellia sinensis)* Plus Caffeic and Chlorogenic Acids, as Shown via DFT and Hydrodynamic Voltammetry

**DOI:** 10.3390/ijms25179342

**Published:** 2024-08-28

**Authors:** Francesco Caruso, Raiyan Sakib, Stuart Belli, Alessio Caruso, Miriam Rossi

**Affiliations:** 1Department of Chemistry, Vassar College, Poughkeepsie, NY 12604, USA; 2Department of Chemistry and Chemical Biology, Harvard University, 12 Oxford St., Cambridge, MA 02138, USA

**Keywords:** *Ilex paraguariensis*, *Camellia sinensis*, DFT, hydrodynamic voltammetry, superoxide scavenging, caffeic acid, chlorogenic acid

## Abstract

We describe the antioxidant capability of scavenging the superoxide radical of several tea and yerba mate samples using rotating ring–disk electrochemistry (RRDE). We directly measured superoxide concentrations and detected their decrease upon the addition of an antioxidant to the electrochemical cell. We studied two varieties of yerba mate, two varieties of black tea from Bangladesh, a sample of Pu-erh tea from China, and two components, caffeic acid and chlorogenic acid. All of these plant infusions and components showed strong antioxidant activities, virtually annihilating the available superoxide concentration. Using density functional theory (DFT) calculations, we describe a mechanism of superoxide scavenging via caffeic and chlorogenic acids. Superoxide can initially interact at two sites in these acids: the H4 catechol hydrogen ***(a)*** or the acidic proton of the acid ***(b)***. For ***(a)***, caffeic acid needs an additional π–π superoxide radical, which transfers electron density to the ring and forms a HO_2_^−^ anion. A second caffeic acid proton and HO_2_^−^ anion forms H_2_O_2_. Chlorogenic acid acts differently, as the initial approach of superoxide to the catechol moiety ***(a)*** is enough to form the HO_2_^−^ anion. After an additional acidic proton of chlorogenic acid is given to HO_2_^−^, three well-separated compounds arise: (1) a carboxylate moiety, (2) H_2_O_2_, and a (3) chlorogenic acid semiquinone. The latter can capture a second superoxide in a π–π manner, which remains trapped due to the aromatic ring, as for caffeic acid. With enough of both acids and superoxide radicals, the final products are equivalent: H_2_O_2_ plus a complex of the type [X-acid–η–O_2_], X = caffeic, chlorogenic. Chlorogenic acid ***(b**)*** is described by the following reaction: 2 O_2_^•−^ + 2 chlorogenic acid → 2 chlorogenic carboxylate + O_2_ + H_2_O_2_, and so, it acts as a non-enzymatic superoxide dismutase (SOD) mimic, as shown via the product formation of O_2_ plus H_2_O_2_, which is limited due to chlorogenic acid consumption. Caffeic acid ***(b)*** differs from chlorogenic acid, as there is no acidic proton capture via superoxide. In this case, approaching a second superoxide to the H4 polyphenol moiety forms a HO_2_^−^ anion and, later, an H_2_O_2_ molecule upon the transfer of a second caffeic acid proton.

## 1. Introduction

The use of plant components as infusions to extract flavor and nutrients has been known since ancient times, and in diverse cultural practices, it is often associated with health and medicinal purposes [1]. Herbal “teas” are infusions made from other herbs or plants, while “tea” is made from the *Camellia sinensis* L. plant. The Food and Agriculture Organization (FAO) states that tea is the world’s most-consumed drink, after water [2]. Although the origins of tea drinking are found in Chinese legends, the earliest physical evidence of tea intake, thus far, is from approximately 2400 years ago at a royal funerary site in Zoucheng City, Shandong Province, China [3]. Tea production involves different stages of handling tea leaves, with oxidation (fermentation) being one of the most important. The amount of oxidation determines the different categories of tea: non-oxidized (green), oxidized (black), and partially oxidized (oolong). Black tea has a stronger flavor than green or oolong teas [4]. In this paper, we focus only on black teas. Bangladesh is an important global tea-producing country, and a large amount of its tea production is found in the Sylhet region [5]. Also, an ancient method for storing tea leaves is to compress them into cakes, allowing them to undergo a unique and long (over many years) fermentation. The highly prized tea Pu-erh is produced mostly in the Yunnan province of China [6].

A common hot beverage used in South America is yerba mate, which arises from steeping the leaves and stems of a native plant found in Argentina, Uruguay, Brazil, and Paraguay, *Ilex paraguariensis* [7]. First used in Paraguay by indigenous populations, yerba mate drinking is now common in Argentina, Brazil, Uruguay, and Chile. Its use is also widespread in Middle Eastern countries such as Lebanon and Syria, popularized by South American immigrants after returning to their places of origin. Commercial yerba mate production involves roasting, drying, and maturing the crushed leaves and tiny stems for at least one year [7]. The customary manner of serving yerba mate is steeping the dried plant mixture in hot water, around 70 °C, to make an infusion that is traditionally served in a cup made from a dried gourd, also originating from the same area of *Ilex* growth, and the beverage is drunk through a silver filtering straw (*bombilla*). When the drink was served in this way, drinking yerba mate was a social ritual among families and friends in which the mate cup was passed around for everyone to sip from, with no thought about infection. Mate “cocido” is obtained after adding hot water, and it is served in cups like tea. Nowadays, a common way to prepare this beverage is steeping yerba mate “tea” bags in hot water. 

Since there is much interest in identifying natural antioxidants present in commonly consumed foods that can assist the body’s natural defenses in disease states [when excess reactive oxygen species (ROS), under conditions of oxidative stress, are found], we studied these samples to understand their health benefits. In particular, excess ROS, such as superoxide radicals, under the condition of cellular stress have been implicated in aging complications, cancer, and several inflammatory diseases [8,9]. Caffeic acid and chlorogenic acid (an ester of caffeic acid with quinic acid), Figure 1, are two compounds known to have widespread health benefits, and they are present in the above-mentioned plant beverages, as well as in many foods and coffee [10,11,12,13,14,15,16,17,18,19,20,21,22,23,24,25,26]. In fact, organic acids account for around 3% of the dry matter in tea leaves, and they influence properties including the antioxidant ability [11].

A prime objective of this investigation was to measure the superoxide scavenging capability of the infusion samples described above, along with two of their components, caffeic and chlorogenic acids. In addition, we wished to study, using DFT methods, the scavenging mechanism of caffeic and chlorogenic acids, natural weak acids commonly found in foods and beverages. We also were motivated to study their behavior towards superoxide after the results from recent similar investigations in our laboratory on the antioxidant vitamins E and C (a weak acid) [27] showed that vitamin C is the stronger scavenger. Additionally, by donating its acidic proton, vitamin C also helps reconstitute vitamin E, allowing the latter to keep scavenging superoxide as long as vitamin C is available. 

Several experimental in vitro and in vivo studies to determine the antioxidant activity of yerba mate and teas are available in the literature [11,24,25,26,28,29,30,31,32,33]. Reports describing the ability of chlorogenic and caffeic acids to scavenge free radicals also exist [25,26,28,29,30,34,35,36]. However, many of the methods used in the literature to measure the scavenging capacity are *indirect* methods based on the reduction in stable free radicals, e.g., galvinoxyl, DPPH, TEAC, and ORAC assays [37,38].

In this report, we describe the antioxidant capability of scavenging the superoxide radical of several samples (two varieties of yerba mate from Argentina, two varieties of black tea from Bangladesh, and a sample of Chinese Pu-erh tea, as well as the caffeic and chlorogenic acids found in these beverages) using a recently developed cyclovoltammetry method from our lab, rotating ring–disk electrochemistry (RRDE). Using this technique, after bubbling oxygen in an anhydrous DMSO solution in an electrochemical cell, we *directly* measure the superoxide concentration, and we can detect its decrease upon the addition of an antioxidant. This procedure allows us to quantitatively rank an antioxidant’s activity by measuring its scavenging capacity. We show experimentally that all of these studied plant components are excellent superoxide scavengers. We utilize DFT calculations to describe a mechanism for superoxide scavenging by the two phenolic compounds, caffeic and chlorogenic acids. One of our objectives was to determine, through mechanism studies, the role of the acidic proton of these polyphenols. Earlier studies with vitamin C and vitamin E showed that the vitamin C proton restores vitamin E’s molecular structure and enhances antioxidant activity.

## 2. Results and Discussion

From our experimental RRDE results, we observed that the yerba mate and black tea samples, as well as the caffeic and chlorogenic acids found in these beverages, were all effective superoxide scavengers.

Caffeic and chlorogenic acids are important polyphenols with an acidic functional group. As shown recently, there is the possibility of a superoxide dismutase action (Equation (1)) occurring for polyphenols [39].
2 O_2_^•−^ + 2H^+^ → O_2_ + H_2_O_2_(1)

In a polyphenol scavenger, the hydroxyl group is oxidized via the superoxide radical. In fact, the superoxide radical captures the hydroxyl H atom to establish a HO_2_^−^ moiety (the first step in Figure 1, shown later), namely σ scavenging. When a polyphenol semiquinone is formed, the unpaired electron (originally within the superoxide) is captured via the polyphenol aromatic system. In turn, an additional proton, *not from the scavenger*, can react with HO_2_^−^ to produce H_2_O_2_ (one product of the SOD dismutation reaction, Equation (1)). The other SOD reaction product is O_2_, which forms through the π–π oxidation of the second superoxide reagent that transfers its electron to the aromatic ring in Equation (1). Therefore, the chemistry of the superoxide radical allows it to gain one proton and one H atom, to generate H_2_O_2_, or release an electron, to form O_2_. Some examples of natural products that can perform this dismutation reaction, and act as SOD mimics, are isoflavones [40] and galangin [39]. At times, the π–π oxidation of superoxide is thwarted since the O_2_ molecule is not able to escape from the aromatic ring system. This was seen in butein when the (butein)–η–O_2_ complex was established [41]. 

We were interested in identifying the proton source under physiological conditions necessary for Equation (1). In the literature, the vitamin E–vitamin C couple is well described. After vitamin E scavenges superoxide, it is vitamin C that provides the proton needed for the vitamin E semiquinone [27]. Therefore, the carboxylic acid functional group in caffeic and chlorogenic acid was of interest to us so that we could see whether these weak acids functioned in a similar manner to vitamin C.

### 2.1. DFT Scavenging of Superoxide via Caffeic Acid Catechol Moiety

The structure of caffeic acid, Figure 1, was geometrically optimized using the BIOVIA Materials Studio DMoL3 software, which uses density functional theory (DFT) and is shown in Figure 2. Figure 3 shows the initial approach for a σ attack via the superoxide radical, with a Van der Waals separation of 2.60 Å between the reagent atoms H(hydroxyl-4) and one O(superoxide). Upon DFT geometry minimization (Figure 4), the interacting atoms become closer, at 1.299 Å, while the aromatic hydroxyl O–H bond distance slightly elongates to 1.157 Å. However, the DFT reaction with expected products, a HO_2_^−^ anion plus an excluded H4 caffeic acid semiquinone radical, has higher energy, and so there is no proton capture via superoxide for this σ approach. The whole arrangement is an anionic radical charged −1. An option was to approach a second superoxide in the π–π manner. The resulting arrangement is a non-radical anion charged (−2), and Figure 5 shows the initial arrangement with a Van der Waals separation between the centroids of superoxide and the aromatic ring at 3.50 Å. DFT minimization (Figure 6) shows the proton captured via σ-superoxide, with the O(superoxide)–H distance at 1.019 Å, and the formed HO_2_^−^ anion resulting in good separation from the remaining aromatic non-radical moiety at 1.644 Å. Meanwhile, the π-approached superoxide transferred some electron density to the ring as the separation between centroids shortened to 3.342 Å, and the superoxide O–O distance was slightly shortened to 1.323 Å. Next, a proton was placed at van der Waals separation (2.60 Å) near the most exposed HO_2_^−^ oxygen, and DFT minimization showed the formation of both an O–H bond (0.981 Å) and H_2_O_2_. The latter was well separated from the remaining aromatic species at 1.626 Å for this non-radical anionic (−1) arrangement (Figure 7). Meanwhile, the π-superoxide strengthened its interaction with the aromatic ring at 3.141 Å (from 3.342 Å in Figure 6), and the separation between O atoms in the π–π stacked superoxide was shortened (from 1.323 Å to 1.299 Å).

A related calculation, using caffeic acid instead of the proton in Figure 7, was performed, and it confirmed the proton capture (Figure 8). H_2_O_2_ resulted in good separation from both polyphenols and the two O–H bond lengths of 1.631 Å and 1.599 Å in Figure 8. The former is closely related to the 1.626 Å in Figure 7. The remaining caffeic carboxylate entity was well separated from H_2_O_2_ at 1.559 Å. The arrangement shown in Figure 8 is anionic (−2 charge) and non-radical. 

Upon the elimination of H_2_O_2_ and caffeic carboxylate in Figure 8, the remaining caffeic acid derivative anion was minimized, and Figure 9 shows that the π–π separation between the ring and the superoxide centroids was similar, at 3.158 Å, to that shown in Figure 8 of 3.310 Å. Also, the O–O bond of this superoxide was shorter (1.302 Å) than normal, isolated superoxide (1.373 Å). This effect was due to the partial transfer of the superoxide electron density to the aromatic ring, which matched the ring’s subsequent loss in aromaticity. Thus, two short C–C bonds, C2–C3 (1.384 Å) and C5–C6 (1.385 Å), contrasted with the longer C1–C2 (1.434 Å), C3–C4 (1.441 Å), C4–C5 (1.462 Å), and C5–C6 (1.426 Å) bonds, implying extended conjugation with the central C=C bond of 1.437 Å. These differences can be compared with the related C–C bond distances in Figure 2. This agrees with a quinone-like electronic configuration: a C4–O4 partial double-bond length (1.286 Å), which was shorter than the single C5–O5 bond (1.371 Å). When the hydroxyl of a second caffeic acid approached O4, its proton was not captured via the η–O_2_–caffeic anion derivative. In an additional calculation, we attempted to react a proton with the compound shown in Figure 9, and this proton, in contrast, was captured, as shown in Figure 10.

Therefore, the DFT results (Figure 9) demonstrate that a proton, which initially was placed at van der Waals separation (2.60 Å) near O4, was captured, O4-H = 0.979 Å, and the structure was modified, with a C4–O4 single bond (1.365 Å) similar to C5–O5 (1.369 Å). Thus, the formation of a caffeic acid complex with the inserted π–π superoxide (3.037 Å), i.e., η–O_2_–caffeic acid, seems feasible when a stronger acid than caffeic acid is the reagent for this neutral, no-spin compound. According to these DFT results, we conclude that a stronger acid is needed to reform the hydroxyl moiety in position 4. Figure 1 summarizes these reactions.

We further conclude that superoxide was able to interact with the caffeic acid polyphenol proton to form anionic HO_2_^−^ that was separated from the carboxylate product by 1.664 Å (Figure 6), and later, another molecule of caffeic acid was able to provide proton-stabilizing H_2_O_2_. However, no further reactivity of caffeic acid was verified through DFT. In addition, the final step of this process, the incorporation of a proton to form the [caffeic acid–η–O_2_] complex, is not feasible for a second caffeic acid donor, and a stronger acid is needed.

### 2.2. DFT Scavenging of Superoxide via Chlorogenic Acid Catechol Moiety 

Chlorogenic acid is an ester obtained through the formal condensation of trans-caffeic acid with quinic acid. Figure 11, Figure 12, Figure 13, Figure 14, Figure 15 and Figure 16 and Appendix A describe its catechol scavenging of superoxide. Important intramolecular H-bonds involve the carbohydrate moiety, shown in Figure 11. H4 is available to scavenge superoxide, and a transition state is involved (Figure 12 and Figure 13). The [HO_2_ chlorogenic semiquinone] (−1) anion, shown in Figure 12, right, interacted with a second chlorogenic acid so that its acidic proton was placed near the most exposed oxygen atom of HO_2_^−^, 2.60 Å apart. Upon DFT minimization (Figure 14), H_2_O_2_ forms and is well separated from both the remaining species, as shown with distances of 1.641 Å and 1.638 Å. The added chlorogenic HO-C=O moiety becomes an O-C=O carboxylate whose C–O bond distance shortens to 1.279 Å, while the distance of C4–HO_2_^−^ (Figure 14), 1.304 Å, becomes 1.298 Å, e.g., increasing its double-bond character. The whole radical arrangement is monoanionic.

After the elimination of H_2_O_2_, the remaining radical chlorogenic carboxylate fragment (Figure 14, bottom right) was minimized. Next, we included a second π–π superoxide (Appendix A), and upon DFT minimization, the separation between centroids of 3.464 Å became slightly shorter than the corresponding Van der Waals distance, 3.50 Å (Figure 15), describing a very weak π–π interaction for this non-radical, chlorogenic anion derivative. The attempt to add a proton from another chlorogenic acid was not successful, and the final product we propose is η–O_2_–(chlorogenic carboxylate). Finally, the interaction of a proton with Figure 15’s structure showed proton incorporation, η–O_2_–(chlorogenic acid) (Figure 16). We conclude that, as for caffeic acid, superoxide can interact with chlorogenic acid polyphenol hydrogen to form a HO_2_^−^ anion separated from the carboxylate product by 1.585 Å (Figure 12). This can react later with another molecule of chlorogenic acid to provide a proton, stabilizing H_2_O_2_, but no further reactivity of chlorogenic acid was verified with DFT. In addition, the final step of the potential process, the incorporation of a proton to form the [chlorogenic acid–η–O_2_] complex, is not feasible, and a stronger acid is needed (Figure 16). We conclude that, when interacting with superoxide, the catechol moiety of chlorogenic acid behaves similarly to that of caffeic acid.

### 2.3. DFT Scavenging of Superoxide via the Acidic Proton of Chlorogenic Acid

Our first approach was to verify whether a hydroxyl H atom from chlorogenic acid was able to be captured via superoxide. Appendix A shows the initial state for this calculation, and DFT minimization shows the HO_2_^−^ anion well separated from the carboxylate chlorogenic radical at 1.575 Å (Figure 17). Therefore, additional chlorogenic acid was placed at van der Waals distance near the HO_2_^−^ most exposed oxygen, and upon DFT minimization, the original superoxide moiety became equally separated from both chlorogenic acid protons, but H_2_O_2_ did not form, as the superoxide was separated from both protons at 1.385 Å and 1.377 Å (Figure 18).

Next, the interaction between a second superoxide and one aromatic ring of Figure 18’s conformation was explored. The DFT of this anionic (-2), non-radical arrangement (Figure 19) resulted in a dramatic conformational change. In fact, H_2_O_2_ formed; it was well separated from both carboxylate moieties of chlorogenic acid at 1.577 Å and 1.580 Å, and the π–π second superoxide, left, became an O_2_ molecule. The stabilization of carboxylate moieties was helped by H-bonds of 1.866 Å and 1.876 Å. The net reaction for the acidic proton of chlorogenic acid is shown in Equation (2).
2 O_2_^•−^ + 2 chlorogenic acid → 2 chlorogenic carboxylate + O_2_ + H_2_O_2_(2)

This may potentially be associated with a non-enzymatic SOD action through which two protons are provided via chlorogenic acid.

### 2.4. DFT Scavenging of Superoxide via the Acidic Proton of Caffeic Acid 

A σ approach is shown in Appendix A, according to a 2.60-Å separation between reacting atoms. The related DFT minimization is shown in Figure 20, where potentially H- and O-reacting atoms are separated by 1.648 Å. The potential scavenged superoxide should form a HO_2_^−^ anionic moiety, and when the arrangement reaction product (including a HO_2_^−^ anion separated at van der Waals distance and a proton-excluded caffeic radical) was DFT-minimized, the HO_2_^−^ proton was recaptured via caffeic acid, establishing the same minimum shown in Figure 20. Thus, there was no capture of the acidic proton via superoxide. Next, a second superoxide was placed near H4 (Appendix A), and DFT minimization showed the formation of the corresponding HO_2_^−^ moiety via O4 (Figure 21), confirming no capture of the acidic proton. The alternative reaction of Figure 20’s arrangement, e.g., a second π–π superoxide approaching the aromatic ring (Appendix A), was DFT-minimized, and this superoxide was redirected towards H4.

We conclude that, although two caffeic H atoms (the hydroxyl H4 and the acidic proton) could interact with superoxide, only the former is successful in scavenging. The following steps in the mechanism of this section are associated with Figure 6 within Figure 1, which was shown earlier.

### 2.5. Experimental Superoxide Measurement Using RRDE to Scavenge the Title Compounds

A straightforward method for generating the superoxide radical is provided via classical cyclic voltammetry in a voltaic cell. Increasing the concentration of antioxidants in the voltaic cell detects the decrease in the current intensity of the superoxide signal [42]. However, the differences among several voltammograms, obtained after successive additions of a scavenger, are problematic to distinguish, and obtaining a quantitative amount of superoxide is challenging. An experimental method, recently improved in our lab [43] by splitting the reduction and oxidation using a rotating ring–disk electrode (RRDE), utilizes hydrodynamic voltammetry to measure the scavenging activity of the title compounds regarding the superoxide radical. At the disk, a reduction produces superoxide following reaction (3), and an example is shown at the bottom of Figure 22, while at the ring, the opposite oxidation reaction is performed following reaction (4) (Figure 22, top). This RRDE method measures superoxide scavenging *directly,* and the superoxide concentration is found. A recent review describes several natural polyphenols analyzed using this RRDE technique [39]. Another review describes details of electroanalysis via antioxidants [44]:Disk electrode: O_2_ + e− → O_2_^•−^(3)
Ring electrode: O_2_^• −^ → O_2_ + e−(4)

We display the experimental voltammograms for caffeic acid in Figure 22, as well as the corresponding collection efficiency in Figure 23. Appendix A includes only the initial five data points collected in Figure 23, showing a linear behavior of y = −11.2457x + 18,434. Its slope, −11.2 × 10^4^, is assigned to the antioxidant capability of scavenging superoxide via caffeic acid. Voltammograms for, and collection efficiency of, chlorogenic acid are shown in Appendix A for black tea, Grade B, Isphahani in Appendix A, and those for black tea, Grade A (Red Label), are shown in Appendix A. Figure 24 shows voltammograms for Pu-erh tea, and its collection efficiency is shown in Appendix A. RRDE data for yerba mate (leaves and stems) are shown in Appendix A; voltammograms for yerba mate (leaves) are shown in Figure 25, and its collection efficiency is depicted in Appendix A.

The measured antioxidant activity of the black tea and yerba mate samples studied showed minor differences, and the slope of the efficiency curve (associated with the scavenging capacity) descended in the following order: black tea, Pu-erh, −0.032 > yerba mate (leaves), −0.024 > black tea, Grade B, Dolia Tea Estate, −0.023 > yerba mate (stems + leaves), −0.018 > black tea, Grade A, Dolia Tea Estate, −0.012. Of the three tea samples, the Pu-erh tea had the strongest activity, suggesting that its unique storage and fermentation method is effective in augmenting its antioxidant activity. The yerba mate slopes were −0.024 (leaves) and −0.018 (stems + leaves), suggesting that the antioxidant compounds of *Ilex paraguariensis* are more concentrated in the leaves than in the stems. This feature coincides with the stronger taste and beverage color when serving the leaves of yerba mate made in the classical way (using a bombilla) and also using the higher-temperature “mate cocido” serving method. Thus, black teas and yerba mate are effective sources of natural antioxidants, especially since they effectively eliminated the superoxide concentration with about 1 mL of solution, as shown in the related voltammograms. 

Experimental RRDE measurements of the two single compounds contained in these plant-derived drinks, caffeic acid, and chlorogenic acid, showed them to be almost equivalent in their ability to scavenge superoxide with slopes of −11.2 × 10^4^ (caffeic acid) and −11.8 × 10^4^ (chlorogenic acid). The activities of these two acidic polyphenols are ranked between butein (−11.2 × 10^4^) [41] and clovamide (−12.0 × 10^4^) [45]. Moreover, when comparing these four polyphenols that have a catechol moiety with non-catecholic polyphenols, for instance, chrysin (slope: −1.10 × 10^4^, one order of magnitude weaker in scavenging) [39], it is seen that the catechol fragment markedly increases the scavenging activity. Interestingly, the commercial antioxidant butylatedhydroxytoluene (BHT) (also a non-catecholic polyphenol) is even two orders of magnitude weaker as a scavenger (−0.16 × 10^4^) [46]. 

A comparison of the RRDE collection efficiency of tea and yerba mate with that of other natural products, also analyzed through the RRDE method, is limited because the aliquots along the *x*-axis of collection efficiency are not of the added single-molecule concentrations but, rather, the volume of the added natural product. In any case, the slopes can indicate trends, and it is useful to compare them. The strongest antioxidant scavenging of superoxide comes from neem oil (slope: −2.271) [47], which contains a large number of antioxidant compounds. Other examples are olive oil (slope: −0.084) [48], black seed oil (slope: −0.078) [49], and bee propolis (slope: −0.086) [50]. That is, with the exception of neem oil, all are scavengers in the −10^−2^-slope range, which coincides with the present study for tea and yerba mate samples.

## 3. Materials and Methods

### 3.1. Materials for RRDE Solutions

Tetrabutylammonium bromide, TBAB (Sigma-Aldrich, St. Louis, MO, USA); dimethyl sulfoxide, DMSO, anhydrous, ≥99.9% (Sigma-Aldrich, St. Louis, MO, USA). 

Caffeic acid (Sigma-Aldrich, St. Louis, MO, USA). RRDE solution: concentration of 0.03 M (0.054 g in 10 mL DMSO), clear, light yellow.

Chlorogenic acid (Sigma Aldrich, St. Louis, MO, USA). RRDE solution: concentration of 0.03 M (0.1063 g in 10 mL DMSO), clear, colorless.

Pu-erh tea. Company: Pu Erh Cheng Shang Wen Hua Chan Ye. Less fermented, rawer Pu-erh leaves; leaves plucked in 2014 from Yunnan Province. Sample was from tea-cake packaged in 2018. RRDE solution concentration: 0.4706 g in 10 mL of DMSO, (51 °C-temp stirring for 10 min), dark brown.

Yerba mate tea, La Merced de Monte, Establecimiento Las Marias, Corrientes, Argentina (leaves only). RRDE solution concentration: 0.4702 g in 10 mL of DMSO (51 °C-temp stirring for 10 min), light green.

Yerba mate tea, Rosamonte, Hreñuk S.A., Misiones, Argentina (leaves and stems). RRDE solution concentration: 0.4376 g in 10 mL of DMSO (51 °C-temp stirring for 10 min), dark brown.

Black tea, Grade A, Dolia Tea Estate, Sylhet, Bangladesh. RRDE solution concentration: 0.4715 g in 10 mL of DMSO (51 °C-temp stirring for 10 min), dark brown.

Black tea, Grade B, Dolia Tea Estate, Sylhet, Bangladesh. RRDE solution concentration: 0.4706 g in 10 mL of DMSO (51 °C-temp stirring for 10 min), dark brown.

### 3.2. Hydrodynamic Voltammetry (RRDE)

Hydrodynamic voltammetry was performed using a rotating ring–disk electrode (RRDE) and the WaveDriver 20 bipotentiostat (Pine Research, Durham, NC, USA) with the MSR Electrode Rotator, as well as CE and ETL marks, also from Pine Research. The electrode tip used was an E6R1 ChangeDisk with a fixed gold ring and gold disk insert (Pine Research). A 0.05-µg alumina suspension (Pine Research) was used to polish the Au/Au working electrode tip in order to clear potential film formation before use. 

In RRDE, the bipotentiostat measures were obtained at the same time using both the currents at the disk and ring electrodes (that correspond to charge movements among the ring, the disk, and the counter-electrode) and the potentials of the disk and ring electrodes relating to the single reference electrode. The 5-neck electrochemical RRDE cell contained four electrodes—the two gold rotating ring and disk working electrodes (Pine Research), one coiled platinum wire counter-electrode, and one reference electrode consisting of a platinum wire. These were placed in a solution containing 0.1 M (1.61 g) of dried TBAB dissolved in 50 mL of dimethyl sulfoxide in a fritted glass tube. The electrochemical cell had the means of either bubbling or blanketing the solution with gas. The solution was subsequently bubbled with dry O_2_/N_2_ (35%/65%) for at least five minutes to establish the required dissolved molecular oxygen level. The partial oxygen-mixture gas tank was used to allow oxygen to flow into the voltaic cell at a comparable rate to that at which the antioxidant was scavenging the generated superoxide radical. 

The Aftermath software release 1.6.10523 (Pine Research Instrumentation, Durham, NC, USA) was used to set up the parameters needed for the experiment: the potential sweep was applied to the disk from 0.2 V to −1.2 V and then reversed to 0.2 V, while the potential of the ring electrode was held constant at 0.0 V. The disk potential was set to sweep at set intervals in the negative direction in order to reduce the superoxide molecules that reacted at the ring to oxygen molecules. The disk’s voltage sweep rate was set to 25 mV/s. The rotation setting used for the rotation of the Au/Au disk electrode was chosen to be 1000 rpm at the disk electrode. The superoxide radical was generated through a molecular oxygen reduction, and the peak was detected at around −0.6 V. Meanwhile, the reverse-oxidation reaction of the remaining unreacted superoxide radicals was detected at the ring electrode.

An initial blank solution consisting of bubbled O_2_, the electrolyte TBMB, and DMSO was run, and the voltammogram showing the current vs. potential graphs was recorded using the Aftermath software (Pine Research). The ratio of the ring–disk current was defined as “efficiency”. Next, an initial antioxidant aliquot was introduced. The solution in the voltaic cell was bubbled with the gas mixture for 5 min, a new voltammogram was recorded, and the corresponding efficiency was calculated. In this way, the rate at which the increasing concentration of the antioxidant scavenged the generated superoxide radicals during the electrochemical reaction was determined upon the addition of each antioxidant aliquot. These data were later evaluated using Microsoft Excel. The volume amount used in each of the aliquots is indicated in the related RRDE graph. Finally, the decreasing slope of the curve, describing the overall decrease in efficiency upon the incremental addition of the antioxidant, serves as a quantitative measure of the antioxidant activity of the samples. Any decrease in the collection efficiency was anticipated to be due to the amount of superoxide consumed via the antioxidant. The methodological development of this procedure was conducted at our laboratory [43]. 

### 3.3. Computational Study

Calculations were run using BIOVIA Materials Studio DMoL^3^, implemented in Materials Studio version 7.0. DMoL^3^ is a modeling program (Dassault Systèmes, San Diego, CA, USA) that utilizes density functional theory (DFT) to calculate properties of molecules such as energy, geometry, and transition-state optimizations [51]. The results allow the relationships between a molecular structure and its antioxidant properties and scavenging behavior to become evident. We employed the double numerical polarized (DNP) basis set, including all the occupied atomic orbitals, plus a second set of valence atomic orbitals, as well as polarized d-valence orbitals [52]. A correlation generalized gradient approximation (GGA) was used, including the BLYP correlation and Becke exchange [53]. In addition, Grimme’s correction was applied when Van der Waals interactions were involved [54]. The effect of the solvent, DMSO, was included in these calculations to better mimic the RRDE environment, and also in the water solvent using the continuous model of Dmol^3^ [55], no important structural differences were observed. All electrons were treated explicitly, and the real space cutoff of 5 Å was set for the numerical integration of the Hamiltonian matrix elements. The self-consistent field convergence criterion was established for the root mean square variation in the electronic density at less than 10^−6^ electron/Å^3^. The convergence criteria applied during geometry optimization were 2.72 × 10^−4^ eV for energy and 0.054 eV/Å for force.

## 4. Conclusions

Varieties of yerba mate and black tea, as well as two components, caffeic acid and chlorogenic acid, show the marked scavenging of the superoxide radical using experimental cyclovoltammetry, and this scavenging is even able to virtually annihilate the available superoxide concentration. Thus, black tea and yerba mate are effective sources of natural antioxidants. In this study, using DFT methods, we also described a mechanism of superoxide scavenging via caffeic and chlorogenic acids. Caffeic acid needs to interact with two superoxide radicals to establish the formation of a HO_2_^−^ anion: one superoxide approaches the catechol hydrogen atom (H4) of caffeic acid, and the second stacks with the aromatic ring. The π–π superoxide transfers electron density to the ring and remains trapped in this mechanism. The interaction between the acidic proton of a second caffeic acid and HO_2_^−^ anion can form H_2_O_2_. For chlorogenic acid, the catechol H4 is also available to scavenge superoxide in a closely related manner to that of caffeic acid. On the other hand, the approach of the carboxylic acid moiety in caffeic acid and chlorogenic acid towards superoxide is different. In chlorogenic acid, three well-separated compounds are calculated via DFT: H_2_O_2_ plus two chlorogenic carboxylates. In addition, there is a release of the unpaired electron via the π–π superoxide and the formation of a molecule of O_2_ that leaves the aromatic ring. This is in marked contrast with chlorogenic acid, as the scavenging of superoxide through its acidic proton shows no formation of a molecule of O_2_.

In summary, according to our DFT results, superoxide scavenging via caffeic and chlorogenic acids determines the formation of HO_2_^−^, which reacts later with a proton released via the acids to form H_2_O_2_. This supports the scavenging mechanism described previously, which shows that added protons were able to perform such a role [27], and this mechanism warrants further investigation. We conclude that caffeic and chlorogenic acid are strong antioxidants, and we confirm our hypothesis that a polyphenol containing an acidic functional group can play an important role in scavenging superoxide. The combination of experimental superoxide scavenging studies using electrochemical methods, along with theoretical calculations to identify a related mechanism, provides valuable information. Further related studies on other beverages are currently underway at our laboratory.

## Data Availability

Data supporting reported results can be obtained by writing to authors.

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
