# Peer review of "Antioxidant Scavenging of the Superoxide Radical by Yerba Mate (Ilex paraguariensis) and Black Tea (Camellia sinensis) Plus Caffeic and Chlorogenic Acids, as Shown via DFT and Hydrodynamic Voltammetry"

_ijms, 2024, doi:10.3390/ijms25179342_

Round 1

Reviewer 1 Report

Comments and Suggestions for Authors

 This study evaluates the antioxidant capabilities of various tea and yerba mate samples using Rotating Ring Disk Electrochemistry (RRDE) to assess their superoxide radical scavenging effectiveness. Analysis includes yerba mate, black tea from Bangladesh, Pu-erh tea from China, caffeic acid, and chlorogenic acid. All samples demonstrated strong antioxidant activity, effectively reducing superoxide levels. Density Functional Theory (DFT) calculations reveal distinct scavenging mechanisms: caffeic acid requires two interactions with superoxide radicals for effective scavenging, whereas chlorogenic acid only needs one. Chlorogenic acid also mimics superoxide dismutase activity, forming O2 and H2O2.

This research well describes the mechanisms of antioxidant capabilities of various tea leaves and antioxidant substances. 

However, several points should be edited.

1 abstract is too long and should be shortened.

2 in the results and discussion section, short subtitles should be described.

3 figure titles should describe the substance analyzed.

4 figure legends should be added

Comments on the Quality of English Language

none

Author Response

Comments and Suggestions for Authors

 This study evaluates the antioxidant capabilities of various tea and yerba mate samples using Rotating Ring Disk Electrochemistry (RRDE) to assess their superoxide radical scavenging effectiveness. Analysis includes yerba mate, black tea from Bangladesh, Pu-erh tea from China, caffeic acid, and chlorogenic acid. All samples demonstrated strong antioxidant activity, effectively reducing superoxide levels. Density Functional Theory (DFT) calculations reveal distinct scavenging mechanisms: caffeic acid requires two interactions with superoxide radicals for effective scavenging, whereas chlorogenic acid only needs one. Chlorogenic acid also mimics superoxide dismutase activity, forming O2 and H2O2.

This research well describes the mechanisms of antioxidant capabilities of various tea leaves and antioxidant substances. 

However, several points should be edited.

1 abstract is too long and should be shortened.

   R:   Done

2 in the results and discussion section, short subtitles should be described.

  R:  We believe it is fine as it is

3 figure titles should describe the substance analyzed.

   R:    Each figure caption describes the DFT process associated to the corresponding figure

4 figure legends should be added

  Figure captions are provided

Reviewer 2 Report

Comments and Suggestions for Authors

Antioxidant scavenging of the superoxide radical by Yerba Mate (Ilex paraguariensis) and Black Tea (Camellia sinensis) plus Caffeic and Chlorogenic acids, as shown by DFT and Hydrodynamic Voltammetry”.

Abstract:

The abstract is too long. The text needs to be summarized and reduced.

“Using DFT calculations we describe a mechanism of superoxide scavenging by caffeic and chlorogenic acids.”

Don’t use undefined abbreviations.

The Abstract is an introduction and not a summary of why the work was carried out, the novelty of the research and the main advances achieved in the research work.

Introduction:

“The use of plant components (leaves, fruits, twigs, etc.) as infusions to extract the

flavor and nutrients is known since ancient times and in diverse cultural practices, often

associated with health and medicinal purposes [1]. Herbal "teas" are infusions made from

other herbs or plants while “tea’ is made from the Camellia sinensis L. plant. The Food and

Agriculture Organization (FAO) states tea is the world’s most consumed drink, after water

[2]. Although the origins of tea drinking are found in Chinese legends, the earliest physical

evidence, thus far, for tea intake is from approximately 2400 years ago at a funerary site

from a royal tomb in Zoucheng City, Shandong Province, China [3].

Tea production involves different stages of handling the tea leaves with oxidation

(fermentation) being one of the most important. The amount of oxidation determines the

different categories of tea: non-oxidized (green), oxidized (black) and partially oxidized,

(oolong). Black tea has a stronger flavor than green or oolong teas. [4] In this paper, we

focus only on black teas. Bangladesh is an important global tea producing country and a

large amount of its tea production is found in the Sylhet region [5]. Also, an ancient

method for storing tea leaves is to compress them into cakes allowing the tea to undergo

a unique and long fermentation. The resulting tea, known as Pu-erh. produced mostly in

the Yunnan province of China, is highly prized. [6]

A common hot beverage used in South America is yerba mate which arises from

steeping leaves and stems of a native plant found in Argentina, Uruguay, Brazil and Par-

aguay, Ilex paraguariensis [7]. First used in Paraguay by indigenous populations, yerba

mate drinking is now common in Argentina, Brazil, Uruguay and Chile. Its use is also

widespread in Middle Eastern countries such as Lebanon and Syria, popularized by South

American immigrants after returning to their places of origin. Commercial yerba mate

production involves roasting, drying, and maturing the crushed leaves and tiny stems for

at least one year [7]. The customary manner of serving yerba mate is by steeping the dried

plant mixture in hot water, around 70° C, to make an infusion that is traditionally served

in a cup made from a dried gourd, also originating from the same area of Ilex growth, and

drunk through a silver filtering straw (bombilla). Served in this way, drinking yerba mate

was a social ritual among families and friends, where the mate cup was passed around for

everyone to sip, with no thought about infection. Mate “cocido” is obtained after adding

hot water and it is served in cups like tea. Nowadays, a common way to prepare this bev-

erage is by steeping yerba mate “tea” bags in hot water”

It is necessary to rewrite this text, it is well known how tea and yerba mate infusions are prepared, and it is not of scientific interest.

Figure 1 is unnecessary..

“In this report, we describe the antioxidant scavenging capability of several samples

(two varieties of yerba mate from Argentina, two varieties of black tea from Bangladesh,

a sample of Chinese Pu-erh tea as well as caffeic and chlorogenic acids found in these

beverages) for the superoxide radical using a recently developed cyclovoltammetry

method from our lab, the Rotating Ring Disk Electrochemistry (RRDE)”

It is necessary to reference and justify the use of the method and highlight the novelty of this research.

Materials and Methods

1.1   Materials for RRDE Solutions

The extraction process used with Tea and Yerba Mate needs to be better described.

How many tests were performed?

What extraction reactor was used and how were the extraction conditions guaranteed?

Was the moisture content of the samples evaluated? How was this done?

How was the solid fraction separated from the liquid?

No conclusions can be drawn without a well-performed extraction process.

Results and Discussion:

To evaluate the viability of the method is necessary to establish a relationship with other well-known methods

Author Response

Comments and Suggestions for Authors

“Antioxidant scavenging of the superoxide radical by Yerba Mate (Ilex paraguariensis) and Black Tea (Camellia sinensis) plus Caffeic and Chlorogenic acids, as shown by DFT and Hydrodynamic Voltammetry”.

Abstract:

The abstract is too long. The text needs to be summarized and reduced.  

R: Done

“Using DFT calculations we describe a mechanism of superoxide scavenging by caffeic and chlorogenic acids.”

Don’t use undefined abbreviations.

R: DFT is now clarified in Abstract

The Abstract is an introduction and not a summary of why the work was carried out, the novelty of the research and the main advances achieved in the research work.

R: Abstract is now modified

Introduction:

“The use of plant components (leaves, fruits, twigs, etc.) as infusions to extract the

flavor and nutrients is known since ancient times and in diverse cultural practices, often

associated with health and medicinal purposes [1]. Herbal "teas" are infusions made from

other herbs or plants while “tea’ is made from the Camellia sinensis L. plant. The Food and

Agriculture Organization (FAO) states tea is the world’s most consumed drink, after water

[2]. Although the origins of tea drinking are found in Chinese legends, the earliest physical

evidence, thus far, for tea intake is from approximately 2400 years ago at a funerary site

from a royal tomb in Zoucheng City, Shandong Province, China [3].

Tea production involves different stages of handling the tea leaves with oxidation

(fermentation) being one of the most important. The amount of oxidation determines the

different categories of tea: non-oxidized (green), oxidized (black) and partially oxidized,

(oolong). Black tea has a stronger flavor than green or oolong teas. [4] In this paper, we

focus only on black teas. Bangladesh is an important global tea producing country and a

large amount of its tea production is found in the Sylhet region [5]. Also, an ancient

method for storing tea leaves is to compress them into cakes allowing the tea to undergo

a unique and long fermentation. The resulting tea, known as Pu-erh. produced mostly in

the Yunnan province of China, is highly prized. [6]

A common hot beverage used in South America is yerba mate which arises from

steeping leaves and stems of a native plant found in Argentina, Uruguay, Brazil and Par-

aguay, Ilex paraguariensis [7]. First used in Paraguay by indigenous populations, yerba

mate drinking is now common in Argentina, Brazil, Uruguay and Chile. Its use is also

widespread in Middle Eastern countries such as Lebanon and Syria, popularized by South

American immigrants after returning to their places of origin. Commercial yerba mate

production involves roasting, drying, and maturing the crushed leaves and tiny stems for

at least one year [7]. The customary manner of serving yerba mate is by steeping the dried

plant mixture in hot water, around 70° C, to make an infusion that is traditionally served

in a cup made from a dried gourd, also originating from the same area of Ilex growth, and

drunk through a silver filtering straw (bombilla). Served in this way, drinking yerba mate

was a social ritual among families and friends, where the mate cup was passed around for

everyone to sip, with no thought about infection. Mate “cocido” is obtained after adding

hot water and it is served in cups like tea. Nowadays, a common way to prepare this bev-

erage is by steeping yerba mate “tea” bags in hot water”

It is necessary to rewrite this text, it is well known how tea and yerba mate infusions are prepared, and it is not of scientific interest.

R: Text is updated

Figure 1 is unnecessary.

R: DFT figures are in several styles that do not shows double bonds. The canonical structures are a guide for the readers.

“In this report, we describe the antioxidant scavenging capability of several samples

(two varieties of yerba mate from Argentina, two varieties of black tea from Bangladesh,

a sample of Chinese Pu-erh tea as well as caffeic and chlorogenic acids found in these

beverages) for the superoxide radical using a recently developed cyclovoltammetry

method from our lab, the Rotating Ring Disk Electrochemistry (RRDE)”

It is necessary to reference and justify the use of the method and highlight the novelty of this research.

R: This is done in section 2.5

Materials and Methods

1.1   Materials for RRDE Solutions

The extraction process used with Tea and Yerba Mate needs to be better described.

How many tests were performed?

R: Several years ago we verified the consistency of our results repeating the experiment. In all previous published works we provided only one test.

What extraction reactor was used and how were the extraction conditions guaranteed?

R: The extraction process is performed by adding DMSO to the solids in a volumetric tube, 10 ml, and further agitation. As shown by extra virgin olive oil, the cold pressure extraction of olives save the antioxidants. Thus, due to sensitive to heating our experiments are performed at room T to preserve the antioxidants. The solid residual was decanted by centrifugation and the aliquots were taken from the supernatant.

Was the moisture content of the samples evaluated? How was this done?

R: We performed the study avoiding humidity using DMSO anhydrous grade.

How was the solid fraction separated from the liquid?

R: It was just left at the bottom of the tube, only the solution was used in the experiment

No conclusions can be drawn without a well-performed extraction process.

Results and Discussion:

To evaluate the viability of the method is necessary to establish a relationship with other well-known methods

R: Our method is an improvement of the that described in Ref [48]. A more deeply explanation was provided by us in our first study using our method [39]. The advantage of our method is the straightforward determination of superoxide concentration, that is, how superoxide decreases after adding a scavenger.

Submission Date

22 July 2024

Reviewer 3 Report

Comments and Suggestions for Authors

Manuscript titled: "Antioxidant scavenging of the superoxide radical by Yerba..." by Francesco Caruso is an interesting and innovative work. I believe that after a few minor revisions, it is definitely suitable for publication in IJMS.

Here are my minor comments:

  1. The title is too long and dull – please make it more concise.
  2. The abstract also seems too long considering the journal’s guidelines.
  3. The first sentence in the abstract is rather unnecessary – it’s more appropriate for the introduction.
  4. Are these keywords: "caffeic acid; chlorogenic acid" necessary?
  5. "The use of plant components (leaves, fruits, twigs, etc.) as infusions to extract..." This mirrors the first sentence of the abstract, so why repeat it?
  6. "The resulting tea, known as Pu-erh. produced mostly in the Yunnan province of China, is highly prized" – What is the point of this sentence? It’s well-known that tea is highly prized.
  7. "The resulting experimental findings do not always reflect the physiological conditions under which..." – What does this sentence mean? Please clarify or remove it.
  8. "One of the questions we wished to clarify in the mechanism studies was to determine if the acidic part of these polyphenols can scavenge superoxide radicals synergistically, as earlier seen in vitamin C and vitamin E studies." – A sentence or two of explanation would be helpful.
  9. BIOVIA Materials Studio DMoL – Why this program and not Gaussian? What is its advantage over Gaussian?
  10. Why write "Equation 1" – isn’t it better to simply put (1)?
  11. Figure 13 – might be better in the Supplementary Materials (SM)? Besides, it is extremely unclear.
  12. Figure S1. The... – Figure S1 in the manuscript? This is strange, why not in a separate file? It causes confusion and mess.
  13. Figures 22 and 23 – firstly, there is a lot of dead space, and secondly, why not make it SM?
  14. Figure S16 – I believe you understand what I mean.
  15. Figure 25 – this is over the top.

In summary, a very interesting article, there is hardly anything to criticize. After minor revisions, it is definitely suitable for publication in IJMS.

Author Response

Comments and Suggestions for Authors

Manuscript titled: "Antioxidant scavenging of the superoxide radical by Yerba..." by Francesco Caruso is an interesting and innovative work. I believe that after a few minor revisions, it is definitely suitable for publication in IJMS.

Here are my minor comments:

  1. The title is too long and dull – please make it more concise. We believe it is succinct.
  2. The abstract also seems too long considering the journal’s guidelines. OK, it is now shortened
  3. The first sentence in the abstract is rather unnecessary – it’s more appropriate for the introduction. Done
  4. Are these keywords: "caffeic acid; chlorogenic acid" necessary?

These 2 polyphenol scavengers are unique as they include also an acidic function, as indicated in Introduction. We believe they should be included.

  1. "The use of plant components (leaves, fruits, twigs, etc.) as infusions to extract..." This mirrors the first sentence of the abstract, so why repeat it? Done
  2. "The resulting tea, known as Pu-erh. produced mostly in the Yunnan province of China, is highly prized" – What is the point of this sentence? It’s well-known that tea is highly prized. Done. This was modified accordingly
  3. "The resulting experimental findings do not always reflect the physiological conditions under which..." – What does this sentence mean? Please clarify or remove it. It is removed.
  4. "One of the questions we wished to clarify in the mechanism studies was to determine if the acidic part of these polyphenols can scavenge superoxide radicals synergistically, as earlier seen in vitamin C and vitamin E studies." – A sentence or two of explanation would be helpful.

The sentence is now modified. The current study shows each antioxidant (caffeic and chlorogenic acids) can perform superoxide scavenging through its polyphenol moiety as expected, but also, additional scavenging can be done with the acidic proton of a second molecule of these natural acids, (similar to vitamin C). Consequently, these acidic antioxidants do not need a different molecular species (as for the synergistic vitamin E-vitamin C pair).

  1. BIOVIA Materials Studio DMoL – Why this program and not Gaussian? What is its advantage over Gaussian? We currently have the BIOVIA programs at our disposal which is widely used in the scientific community (although it is expensive).
  2. Why write "Equation 1" – isn’t it better to simply put (1)? Done 
  3. Figure 13 – might be better in the Supplementary Materials (SM)? Besides, it is extremely unclear. This is the output of DMOL3 program, It cannot be modified. It shows clearly the barrier and DeltaG. We believe it is illustrative in the manuscript.
  4. Figure S1. The... – Figure S1 in the manuscript? This is strange, why not in a separate file? It causes confusion and mess. OK, Supplementary figures are now placed at the end of the manuscript.
  5. Figures 22 and 23 – firstly, there is a lot of dead space, and secondly, why not make it SM? If the size of Figure 22 is reduced, reading the aliquot information would be very difficult. We reduce now the size of Figure 23.
  6. Figure S16 – I believe you understand what I mean. See 12
  7. Figure 25 – this is over the top. See 12

In summary, a very interesting article, there is hardly anything to criticize. After minor revisions, it is definitely suitable for publication in IJMS.

Thanks for your comments that helped us to improve the manuscript.

Submission Date

22 July 2024

Reviewer 4 Report

Comments and Suggestions for Authors

The work is very interesting. The methodology is very clearly and thoroughly presented. The results are clearly presented, the graphs are properly done and also clear. Recommends that the work can be published in a journal IJMS.

I only ask that you take into account:

figure 1, more readable, would be the notation when one formula is labelled ‘a’ and the other as ‘b’

Author Response

Comments and Suggestions for Authors

The work is very interesting. The methodology is very clearly and thoroughly presented. The results are clearly presented, the graphs are properly done and also clear. Recommends that the work can be published in a journal IJMS.

I only ask that you take into account:

figure 1, more readable, would be the notation when one formula is labelled ‘a’ and the other as ‘b’

The indications “top” and “bottom” clarify both figures

Submission Date

Round 2

Reviewer 2 Report

Comments and Suggestions for Authors

“Antioxidant scavenging of the superoxide radical by Yerba Mate (Ilex paraguariensis) and Black Tea (Camellia sinensis) plus Caffeic and Chlorogenic acids, as shown by DFT and Hydrodynamic Voltammetry”.

Abstract:

The abstract is too long. The text needs to be summarized and reduced.

R: Done

OK

“Using DFT calculations we describe a mechanism of superoxide scavenging by caffeic and chlorogenic acids.”

Don’t use undefined abbreviations.

R: DFT is now clarified in Abstract

OK

The Abstract is an introduction and not a summary of why the work was carried out, the novelty of the research and the main advances achieved in the research work.

R: Abstract is now modified

The abstract is still not a good summary of the work done.

Introduction:

“The use of plant components (leaves, fruits, twigs, etc.) as infusions to extract the

flavor and nutrients is known since ancient times and in diverse cultural practices, often associated with health and medicinal purposes [1]. Herbal "teas" are infusions made from other herbs or plants while “tea’ is made from the Camellia sinensis L. plant. The Food and Agriculture Organization (FAO) states tea is the world’s most consumed drink, after water [2]. Although the origins of tea drinking are found in Chinese legends, the earliest physical evidence, thus far, for tea intake is from approximately 2400 years ago at a funerary site from a royal tomb in Zoucheng City, Shandong Province, China [3].

Tea production involves different stages of handling the tea leaves with oxidation

(fermentation) being one of the most important. The amount of oxidation determines the different categories of tea: non-oxidized (green), oxidized (black) and partially oxidized, (oolong). Black tea has a stronger flavor than green or oolong teas. [4] In this paper, we focus only on black teas. Bangladesh is an important global tea producing country and a large amount of its tea production is found in the Sylhet region [5]. Also, an ancient method for storing tea leaves is to compress them into cakes allowing the tea to undergo a unique and long fermentation. The resulting tea, known as Pu-erh. produced mostly in the Yunnan province of China, is highly prized. [6]

A common hot beverage used in South America is yerba mate which arises from

steeping leaves and stems of a native plant found in Argentina, Uruguay, Brazil and Paraguay, Ilex paraguariensis [7]. First used in Paraguay by indigenous populations, yerba mate drinking is now common in Argentina, Brazil, Uruguay and Chile. Its use is also widespread in Middle Eastern countries such as Lebanon and Syria, popularized by South American immigrants after returning to their places of origin. Commercial yerba mate production involves roasting, drying, and maturing the crushed leaves and tiny stems for at least one year [7]. The customary manner of serving yerba mate is by steeping the dried plant mixture in hot water, around 70° C, to make an infusion that is traditionally served in a cup made from a dried gourd, also originating from the same area of Ilex growth, and drunk through a silver filtering straw (bombilla). Served in this way, drinking yerba mate was a social ritual among families and friends, where the mate cup was passed around for everyone to sip, with no thought about infection. Mate “cocido” is obtained after adding

hot water and it is served in cups like tea. Nowadays, a common way to prepare this beverage is by steeping yerba mate “tea” bags in hot water”

It is necessary to rewrite this text, it is well known how tea and yerba mate infusions are prepared, and it is not of scientific interest.

R: Text is updated

OK

Figure 1 is unnecessary.

R: DFT figures are in several styles that do not shows double bonds. The canonical structures are a guide for the readers.

Figure 1 is unnecessary.

“In this report, we describe the antioxidant scavenging capability of several samples (two varieties of yerba mate from Argentina, two varieties of black tea from Bangladesh, a sample of Chinese Pu-erh tea as well as caffeic and chlorogenic acids found in these beverages) for the superoxide radical using a recently developed cyclovoltammetry method from our lab, the Rotating Ring Disk Electrochemistry (RRDE)”

It is necessary to reference and justify the use of the method and highlight the novelty of this research.

R: This is done in section 2.5

This should be done in the introduction.

Materials and Methods

1.1 Materials for RRDE Solutions

The extraction process used with Tea and Yerba Mate needs to be better described.

How many tests were performed?

R: Several years ago we verified the consistency of our results repeating the experiment. In all previous published works we provided only one test.

Replicates are needed to confirm the results.

What extraction reactor was used and how were the extraction conditions guaranteed?

R: The extraction process is performed by adding DMSO to the solids in a volumetric tube, 10 ml, and further agitation. As shown by extra virgin olive oil, the cold pressure extraction of olives save the antioxidants. Thus, due to sensitive to heating our experiments are performed at room T to preserve the antioxidants. The solid residual was decanted by centrifugation and the aliquots were taken from the supernatant.

The extraction process is still poorly described and seems to have been carried out under poorly controlled conditions. Temperature, solid/liquid ratio, time, stirring intensity, etc. need to be controlled.

Was the moisture content of the samples evaluated? How was this done?

R: We performed the study avoiding humidity using DMSO anhydrous grade.

It is necessary to evaluate the moisture content of Yerba Mate and Tea, in order to determine the solid/liquid ratio used in the extraction process.

How was the solid fraction separated from the liquid?

R: It was just left at the bottom of the tube, only the solution was used in the experiment

And what happened to the liquid retained in the solid fraction?

No conclusions can be drawn without a well-performed extraction process.

The authors do not answer the reviewer's question!!!

Results and Discussion:

To evaluate the viability of the method is necessary to establish a relationship with other well-known methods

R: Our method is an improvement of the that described in Ref [48]. A more deeply explanation was provided by us in our first study using our method [39]. The advantage of our method is the straightforward determination of superoxide concentration, that is, how superoxide decreases after adding a scavenger.

The method was previously tested to evaluate the same extracts. If yes, the work does not present any novelty. If no, it is necessary to contrast the results obtained.

Reviewer 3 Report

Comments and Suggestions for Authors

The authors have made very good revisions to the work, and I believe it can be accepted.

Round 3

Reviewer 2 Report

Comments and Suggestions for Authors

“Antioxidant scavenging of the superoxide radical by Yerba Mate (Ilex paraguariensis) and Black Tea (Camellia sinensis) plus Caffeic and Chlorogenic acids, as shown by DFT and Hydrodynamic Voltammetry”.

Abstract:

The abstract is too long. The text needs to be summarized and reduced.

R: Done

OK

“Using DFT calculations we describe a mechanism of superoxide scavenging by caffeic and chlorogenic acids.”

Don’t use undefined abbreviations.

R: DFT is now clarified in Abstract

OK

The Abstract is an introduction and not a summary of why the work was carried out, the novelty of the research and the main advances achieved in the research work.

R: Abstract is now modified

The abstract is still not a good summary of the work done.

Introduction:

“The use of plant components (leaves, fruits, twigs, etc.) as infusions to extract the

flavor and nutrients is known since ancient times and in diverse cultural practices, often associated with health and medicinal purposes [1]. Herbal "teas" are infusions made from other herbs or plants while “tea’ is made from the Camellia sinensis L. plant. The Food and Agriculture Organization (FAO) states tea is the world’s most consumed drink, after water [2]. Although the origins of tea drinking are found in Chinese legends, the earliest physical evidence, thus far, for tea intake is from approximately 2400 years ago at a funerary site from a royal tomb in Zoucheng City, Shandong Province, China [3].

Tea production involves different stages of handling the tea leaves with oxidation

(fermentation) being one of the most important. The amount of oxidation determines the different categories of tea: non-oxidized (green), oxidized (black) and partially oxidized, (oolong). Black tea has a stronger flavor than green or oolong teas. [4] In this paper, we focus only on black teas. Bangladesh is an important global tea producing country and a large amount of its tea production is found in the Sylhet region [5]. Also, an ancient method for storing tea leaves is to compress them into cakes allowing the tea to undergo a unique and long fermentation. The resulting tea, known as Pu-erh. produced mostly in the Yunnan province of China, is highly prized. [6]

A common hot beverage used in South America is yerba mate which arises from

steeping leaves and stems of a native plant found in Argentina, Uruguay, Brazil and Paraguay, Ilex paraguariensis [7]. First used in Paraguay by indigenous populations, yerba mate drinking is now common in Argentina, Brazil, Uruguay and Chile. Its use is also widespread in Middle Eastern countries such as Lebanon and Syria, popularized by South American immigrants after returning to their places of origin. Commercial yerba mate production involves roasting, drying, and maturing the crushed leaves and tiny stems for at least one year [7]. The customary manner of serving yerba mate is by steeping the dried plant mixture in hot water, around 70° C, to make an infusion that is traditionally served in a cup made from a dried gourd, also originating from the same area of Ilex growth, and drunk through a silver filtering straw (bombilla). Served in this way, drinking yerba mate was a social ritual among families and friends, where the mate cup was passed around for everyone to sip, with no thought about infection. Mate “cocido” is obtained after adding

hot water and it is served in cups like tea. Nowadays, a common way to prepare this beverage is by steeping yerba mate “tea” bags in hot water”

It is necessary to rewrite this text, it is well known how tea and yerba mate infusions are prepared, and it is not of scientific interest.

R: Text is updated

OK

Figure 1 is unnecessary.

R: DFT figures are in several styles that do not shows double bonds. The canonical structures are a guide for the readers.

Figure 1 is unnecessary.

Indeed, as chemists, showing these drawings of chemical structures is important. We believe that a picture showing the canonical structures described in a paper is useful for the reader. We do not believe that this figure is producing any trouble.

As a chemist, figure 1 is unnecessary.

“In this report, we describe the antioxidant scavenging capability of several samples (two varieties of yerba mate from Argentina, two varieties of black tea from Bangladesh, a sample of Chinese Pu-erh tea as well as caffeic and chlorogenic acids found in these beverages) for the superoxide radical using a recently developed cyclovoltammetry method from our lab, the Rotating Ring Disk Electrochemistry (RRDE)”

It is necessary to reference and justify the use of the method and highlight the novelty of this research.

R: This is done in section 2.5

This should be done in the introduction.

Response: In the introduction, we have written “Several experimental in vitro and in vivo studies to determine the antioxidant activity of yerba mate and teas are in the literature [11, 24-26, 28-33]. Reports describing the ability of chlorogenic and caffeic acids to scavenge free radicals also exist [25, 26, 28-30, 34-36]. However, many of the methods used in the literature to measure the scavenging capacity are indirect methods based on the reduction of stable free radicals e.g.galvinoxyl, DPPH, TEAC and ORAC assays [37,38].”

We feel that is sufficient since we go into more detail in section 2.5

The novelty of the method should be better described in the introduction

Materials and Methods

1.1 Materials for RRDE Solutions

The extraction process used with Tea and Yerba Mate needs to be better described.

How many tests were performed?

R: Several years ago we verified the consistency of our results repeating the experiment. In all previous published works we provided only one test.

Replicates are needed to confirm the results.

Extraction procedure: We are not interested in isolating active ingredients through extraction in our experiments, rather we work with the entire plant sample. The extraction is through a simple non-aqueous infusion at room temperature, as described below in the responses, and in the Materials and Methods.

When the authors mix the plant materials with the solvent, they are performing an extraction process that must be reproducible in order to make experiments suitable for publication.

Number of tests performed: We did not replicate the experiments since whenever we checked our experimental results in previous related studies, there were no differences when repeating the experiment. Since our method worked well, we did not want to modify the procedure for this work.

Replicates are needed to confirm the results.

What extraction reactor was used and how were the extraction conditions guaranteed?

R: The extraction process is performed by adding DMSO to the solids in a volumetric tube, 10 ml, and further agitation. As shown by extra virgin olive oil, the cold pressure extraction of olives save the antioxidants. Thus, due to sensitive to heating our experiments are performed at room T to preserve the antioxidants. The solid residual was decanted by centrifugation and the aliquots were taken from the supernatant.

The extraction process is still poorly described and seems to have been carried out under poorly controlled conditions. Temperature, solid/liquid ratio, time, stirring intensity, etc. need to be controlled.

The experiment was performed at room T, in a lab with air T controlled in the building, 21-23 Celsius.

Solid/liquid ratio? In the Materials and Methods section, the weight amount of solid sample, and the volume of DMSO solvent, are well described.

Stirring? It is intrinsic to the hydrodynamic technique. It was performed by rotation of the disk/ring electrode. The contents of the entire electrolytical cell are constantly stirred.

Time? We declared the amount of time of bubbling the O2/N2 mixture for each aliquot experiment (5 minutes). Further details are explained in our first report regarding the method [39].

The extraction process is still poorly described

Was the moisture content of the samples evaluated? How was this done?

R: We performed the study avoiding humidity using DMSO anhydrous grade.

It is necessary to evaluate the moisture content of Yerba Mate and Tea, in order to determine the solid/liquid ratio used in the extraction process.

We hope to achieve extremely low moisture content. Both samples of tea and yerba mate were dry, as received in the commercial samples. We also declared that the gas mixture (O2/N2) was dry and our solvent was dry DMSO. Our intention was to exclude as much water as possible from tea and yerba mate samples and to establish, for these samples, how much they could scavenge superoxide in the absence of water.

As described earlier and in the Materials and Methods section, we used the same amount (weight) of dry sample for the yerba mate and tea samples.

Natural lignocellulosic materials are hygroscopic and despite being dry have an equilibrium moisture content of about 10% in contact with the environment. It is required to evaluate the moisture content

How was the solid fraction separated from the liquid?

R: It was just left at the bottom of the tube, only the solution was used in the experiment

And what happened to the liquid retained in the solid fraction?

Response: Any liquid retained by the solid fraction is left behind. We only use the supernatant.

The solid was not used, as we cannot introduce solid particles in the voltaic cell, since the electrode surfaces are very sensitive to them. That means, only the supernatant liquid solution was used. The aliquots used are very tiny amounts and a minimal part of the tube containing the sample in DMSO.

The process is not replicable

No conclusions can be drawn without a well-performed extraction process.

The authors do not answer the reviewer's question!!!

Response: We are very sorry about this misunderstanding and apologize for any confusion.

However, our method is only to dissolve the plant samples in DMSO (weight:volume), mix thoroughly, and then use only the clear dissolved DMSO plant solution. We arrived at this method development over the course of many previous experiments with different plant samples.

The results of this procedure convinced our group that this was a good method that yields useful results.

The method is not replicable, and the results are not publishable.

Results and Discussion:

To evaluate the viability of the method is necessary to establish a relationship with other well-known methods

R: Our method is an improvement of the that described in Ref [48]. A more deeply explanation was provided by us in our first study using our method [39]. The advantage of our method is the straightforward determination of superoxide concentration, that is, how superoxide decreases after adding a scavenger.

The method was previously tested to evaluate the same extracts. If yes, the work does not present any novelty. If no, it is necessary to contrast the results obtained.

This manuscript describes results of new experiments that were not done before.

As indicated, refs 39 and 48 are good descriptions of the method. More specifically, see section 2.5:

2.5. Experimental superoxide measurement using RRDE for scavenging the

title compounds.

A straightforward method to generate the superoxide radical is provided by classical cyclic voltammetry in a voltaic cell. Increasing the concentration of antioxidants in the voltaic cell detects the decrease of current intensity of superoxide signal [48]. However, the differences among several voltammograms, obtained after successive additions of scavenger are problematic to distinguish

and obtaining a quantitative amount of superoxide is challenging. An experimental method, recently improved in our lab [39] by splitting reduction and oxidation using a rotating ring disk electrode (RRDE), utilizes hydrodynamic voltammetry to measure the scavenging activity of the title compounds regarding the superoxide radical

The unpublished results are the tests carried out with Yerba Mate and Tea extracts? The extraction studies were not well done, and the results do not present the novelty and rigour required for publication.